# A Computer-Assisted Approach Regarding the Optimization of the Geometrical Planning of Medial Opening Wedge High Tibial Osteotomy

**Ileana Ioana Cofaru** [1], **Mihaela Oleksik** [1], **Nicolae Florin Cofaru** [1,*], **Andrei Horia Branescu** [1], **Adrian Haşegan** [2], **Mihai Dan Roman** [2], **Sorin Radu Fleaca** [2] **and Robert Daniel Dobrotă** [3]

1    Faculty of Engineering, "Lucian Blaga" University, 550024 Sibiu, Romania; ioana.cofaru@ulbsibiu.ro (I.I.C.); mihaela.oleksik@ulbsibiu.ro (M.O.); horia.branescu@ulbsibiu.ro (A.H.B.)
2    Faculty of Medicine, "Lucian Blaga" University, 550024 Sibiu, Romania; adrian.hasegan@ulbsibiu.ro (A.H.); mihai.roman@ulbsibiu.ro (M.D.R.); radu.fleaca@ulbsibiu.ro (S.R.F.)
3    Central Military Emergency Hospital Dr. Carol Davila, 010825 Bucharest, Romania; robertdanieldobrota@gmail.com
*    Correspondence: nicolae.cofaru@ulbsibiu.ro

**Abstract:** Opening wedge high tibial osteotomy (OWHTO) is a surgical procedure often used to eliminate the effects of knee osteoarthritis, a disease that is becoming more widespread worldwide. Optimizing the geometric planning of this operation is a very important preparatory step for the success of the intervention and rapid postoperative recovery. This optimization is performed in two main directions. The first direction evaluates the intraoperative behavior of the tibia during the osteotomy by optimizing four geometric parameters that characterize geometric planning. The second direction aims at a postoperative evaluation of the flat tibia-osteosynthesis assembly taking into account the optimal position on the medial–lateral articular line through which the corrected mechanical axis of the tongue passes and implicitly offloads the transfer from the medial area to the side of the knee. The research methods used are exclusively computer-assisted such as: computer-aided design (hereinafter CAD) for geometric modeling of the tibia taking into account the real bone structure, the finite element method (hereinafter FEM) for performing numerical analyses and design of the experiment (hereinafter DOE) for the design of the research. The results obtained are eloquent and clearly presented and can be important elements for orthopedic doctors at the geometric planning stage of the OWHTO.

**Keywords:** opening wedge high tibial osteotomy; geometric planning of the OWHTO; Fujisawa point; TomoFix plate

## 1. Introduction

High tibial osteotomy (hereinafter HTO) is a surgical strategy used successfully to treat osteoarthritis (hereinafter OA) that occurs in the medial compartment of the knee [1–3]. The efficiency of this procedure is given both by the fact that it eliminates the axial deviations that appear at the level of the lower limb as a result of OA and by the fact that it postpones more radical operations, such as total or unicompartmental knee replacements [4]. HTO is preferred for young patients because it allows a relatively quick recovery, with the possibility of a quick return to daily activities, including sports.

As is well known, for the treatment of this condition, which occurs in the medial area of the knee, there are two established types of HTO: opening wedge high tibial osteotomy (hereinafter OWHTO), applied in the medial zone of the knee and closing wedge high tibial osteotomy, hereinafter CWHTO), applied in the lateral zone [2]. Previous research, as well as clinical trials, have shown the superiority of OWHTO as surgery, due to the complications that may occur in the case of the other type of HTO: limb shortening,

neurological problems, detachment of the lateral muscle or proximal fibula osteotomy [5,6]. Even if these complications do not occur in the case of OWHTO, for the success of the intervention and a quick recovery, some aspects must be treated carefully such as: a correct geometric planning of the surgery, establishing a possible overcorrection of the axial deviation caused by OA or finding the best fastener system. All this ensures a good stability of the joint and a fast postoperative recovery. Additionally, some shortcomings are avoided such as: lateral hinge microfractures, loss reduction or instability of the knee joint [7–9].

The need for and importance of the research presented in this article is primarily due to the incidence of OA among adults worldwide [10–14]. Knee osteoarthritis is an increasing condition generating pain and functional disability in people aged over 55 years. Symptomatic knee osteoarthritis generates significant health and economic costs, with almost 85% of the money spent on replacement surgeries [15]. More than this, projections show that this incidence will increase due to the ageing population [16]. Considering the quality of life impairment and the economic impact of knee replacement, the benefits of conservative treatment and knee-preserving surgery should be further analyzed [17].

The biomechanical aspects characterized by significant loads of the tibiofemoral joint are also important, loads that exceed, at certain moments (descending–climbing stairs, jumping, walking, etc.), by several times the body weight [10], and the medial compartment of the knee, which is affected by OA, is more loaded than the lateral one, the ratio being 60% medial and 40% lateral in the case of healthy patients [11–13].

Taking into account the elements presented above, the main purpose of this study is to optimize OWHTO in terms of geometric planning, addressing two main directions.

A first direction concerns the intraoperative behavior of the tibia during the realization of the correction angle; therefore, the creation of the osteotomy wedge. For this, the main geometric parameters that characterize the geometric planning of the OWHTO will be taken into account, namely: the correction angle, the position of the initiation point of the osteotomy plane on the medial cortex of the tibia and the position of the "hinge" relative to the tibial plateau and to the lateral cortex of the tibia. This "hinge" is called the center of the rotation of the angulation (hereinafter, "CORA").

The purpose of this approach is to establish the optimum geometrical planning in order to avoid the microfractures on the lateral cortex of the proximal tibia and on the articular surface of the lateral plateau.

The second direction is a postoperative evaluation of the flat tibia-osteosynthesis assembly. A very important variable that affects the geometric planning of the OWHTO and that we will take into account, is the optimal position on the medial–lateral articular line through which to pass the corrected mechanical axis of the tongue. In the case of healthy subjects, this position is located in the middle of the knee joint, i.e., 50% of the distance measured from the medial to the lateral, and, therefore, normally, this should be the target to achieve after surgery [18].

In reality, in surgical practice, in order to realign the mechanical axis of the foot, the angular correction is not made through the knee joint, but an overcorrection is made. The overcorrections made are practically between 55–75% [18–26], measured in the direction mentioned above, and will have as a consequence a load transfer from the medial to the lateral. From the value point of view, in the paper [27] it is highlighted that, for every 1 mm of transfer of the position of the intersection point of the mechanical axis with the medial–lateral articular line, a load transfer of up to 41 N takes place from the medial compartment to the side.

The reasons for this overcorrection would be the following: biomechanically, anyway, a higher load is taken on the medial (60%) than on the lateral (40%); it also takes into account the increased wear of the medial intraarticular cartilage and the patient's predisposition to develop OA again with the involvement of that area. Under these conditions, a discharge from the medial compartment is welcome.

Many studies recommend overcorrection through a point called the Fujisawa Point located at 62.5% [18,19], measured from the medial to the lateral area. Other approaches

correlate cartilage wear with overcorrection as follows: 55–57.5% for one-third worn cartilage, 60–62.5% for two-thirds worn cartilage, 65–67.5% for the entirely worn cartilage [20]. There are also studies that recommend higher overcorrections: 71.93% [21] or even 75% [22], but these lead to other dysfunctions.

In consequence, another purpose of our study is to establish which is the optimum position of the point on the medial–lateral articular line through which the mechanical axis of the foot will pass after the OWHTO in order to obtain the best biomechanical stability of the joint.

The objectives of the research are the following:

- The realization of a parameterized CAD modeling of the human tibia, as a set of entities with different mechanical characteristics in accordance with the real structure of the tibia.
- The virtual simulation of the OWHTO, with the realization of the "operated tibia" models in different combinations of the parameters that characterize the geometric planning of the OWHTO.
- Modeling of the flat tibia-osteosynthesis assembly after surgery.
- Performing analyses by the finite element method and the Taguchi method to optimize the geometric parameters of the OWHTO.
- Performing finite element analyses applied to the tibia assembly operated by OWHTO and satisfactory TomoFix osteosynthesis, focused on the medial–lateral load transfer due to changes in the positioning of the point of intersection between the medial–lateral articular line and the corrected mechanical axis.

## 2. Materials and Methods

This article is an exclusive computer-aided approach to geometric planning of OWHTO. Consequently, the study and research methods used are computer-aided design (hereinafter CAD), finite element (hereinafter FEM) or design of the experiment (hereinafter DOE).

### 2.1. CAD Modeling of the Tibia Assembly—Plate Tomofix Osteosynthesis after OWHTO Surgery

The use in biomechanical surveys of 3D spatial models of some bones of the human bone system is very appropriate because it allows an understanding and visualization, in detail, of their geometric and dimensional characteristics while allowing the virtual realization of surgery. Finally, and as important, the aim is to perform numerical simulations and analyses using the finite element method of various situations, stresses, loads to which these bones may be subjected [24,28].

In this sense, in this section, a human tibia was modeled in 3D, on which the simulation of the OWHTO operation was performed virtually. Three-dimensional models of the osteosynthesis plate and the fixing screws were also made and, finally, using these 3D models, the 3D assembly of the tibia operated with the applied fixing system was made.

#### 2.1.1. Three-Dimensional Modeling of the Tibia Taking into Account the Real Structures of the Bone

Regarding the 3D modeling of the tibia, there is a trend in the literature for using 3D bone models without taking into account the real heterogeneous structure of human bones [28,29]. Given the use of such models, a number of important research limitations may arise. Therefore, in this research, the aim was to create the CAD model of a real human tibia, constituted as a set of several structural entities with different dimensional, geometric, anatomical and mechanical characteristics.

To achieve this goal, we started from a 3D professional model of a human tibia purchased from the ZYGOTE company, a world leader in 3D anatomical modeling. Figure 1 shows this initial model, highlighting from a dimensional point of view, the total length of the tibia, measured from the tibiotalar joint to the tibial spine (376 mm), its width in the tibial plateau area (78 mm) and the length of the diaphysis, which is 272 mm. Additionally, the three important areas of the bone are presented, namely, distal epiphysis, diaphysis

and proximal epiphysis, respectively. The mentioned dimensions are important, being reference elements in the dimensional characterization of the tibia. Starting from these and using quantitative computed tomography (hereinafter QCT) and images or previous research [30–33], it was possible to make the necessary correlations to model the real entities of which the tibia is constituted.

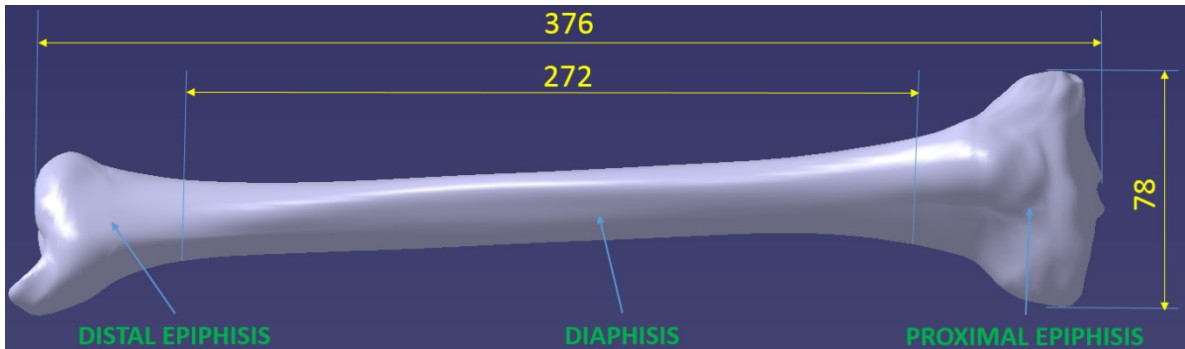

**Figure 1.** Shows the 3D models obtained for the two epiphyses, each of which is the set of two entities corresponding to the areas of cortical and trabecular bone.

As is known [31,32], the 3 components of the tibia mentioned above are structurally different. The diaphysis has a tubular structure made of cortical bone, is compact, with a variable thickness. Inside this structure is the medullary cavity filled with bone marrow. The two epiphyses that form the extremities of the tibia have a heterogeneous bone structure, being formed both of the cortical bone hard to the outside, and of the trabecular or spongy bone inside.

CAD modeling was performed in Catia V5R20 software, taking into account the elements mentioned above.

The main stages of modeling were:

- creating three distinct 3D models from the tibia model: distal epiphysis, diaphysis and proximal epiphysis, respectively, Figure 1;
- remodeling each of these three initial models taking into account the actual bone structure;
- assembling remodeled entities.

A. CAD Modeling of the Epiphysis

Figure 2 shows the 3D models obtained for the two epiphyses, each of which is the set of two entities corresponding to the areas of cortical and trabecular bone.

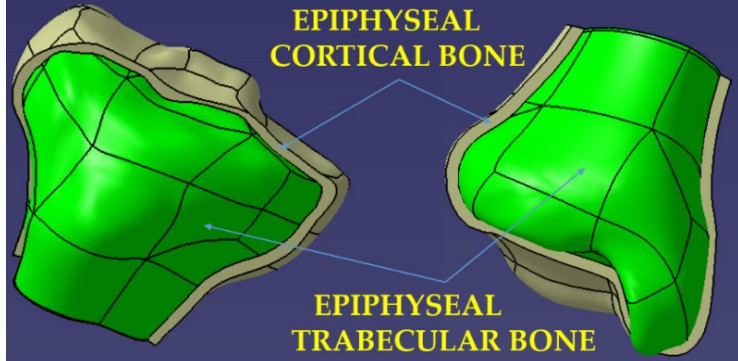

**Figure 2.** Three-dimensional model of the distal and proximal epiphysis—real structure of the bone.

In order to obtain the models for the compact cortical bone area, a spatial off-setting (from the initial model) of the external surfaces of the two epiphyses was performed. The size of the offset, which represents the thickness of the cortical bone at the level of the epiphyses, was adopted of 2.5 mm, a value taken from the literature [30].

For the modeling of the areas formed by the trabecular bone, Boolean operations were used to extract from the initial model of the previously modeled compact cortical bone entities, the extracted part (the green one in Figure 2) thus becoming the 3D model of the trabecular structure of the epiphyses.

B. CAD Modeling of the Diaphysis and Metaphysis

The diaphysis' modeling was more complex due to its variable geometry. In order to obtain the real tubular structure, a sectioning of the initial model of the diaphysis was first performed, which has a length of 272 mm, with 17 planes perpendicular to the tibia, positioned equidistantly. In the case of the real diaphysis, each of these sections is characterized by an outer and an inner contour. Determining these contours and then linking them to CatiaV5R20-specific Solid MultiSection functions makes it possible to obtain the CAD model of the diaphysis entity.

Specifically, the outer contours of each of the 17 sections were first determined using the initial model, resulting in intersections between the sectioned planes corresponding to each section and the body of the diaphysis (Figure 3).

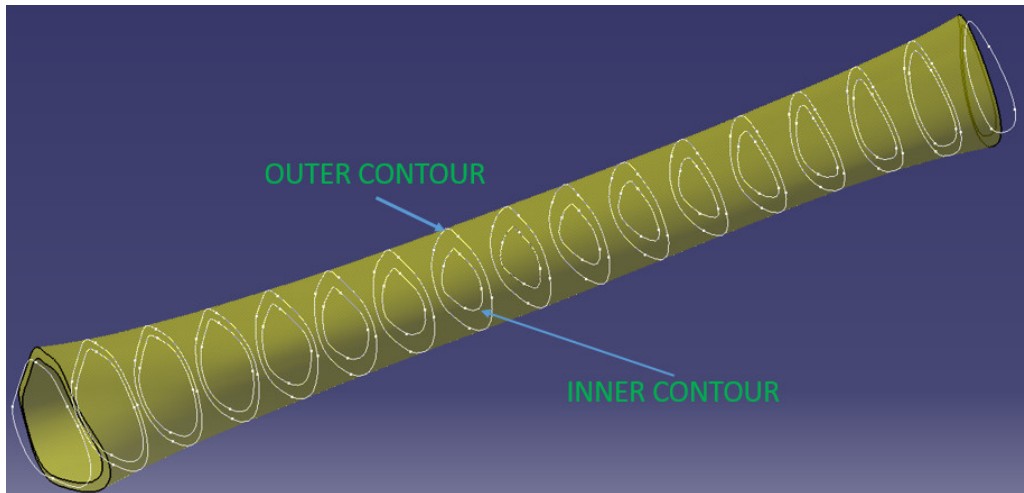

**Figure 3.** Diaphysis 3D model.

To model the inner surface, and the separation between the medullary canal and the area of compact cortical bone, the inner contours of the 17 sections must be defined. The geometry of these contours is variable in both shape and dimensions. In order to be able to make the model, several evaluations with QCT images and work from the specialized literature were studied [30–33], resulting in the following aspects:

- in the areas from the ends of the diaphysis (approximately 1/4–1/3 of its length) the inner contours are equidistant with the outer ones and the offset between them (cortical bone thickness) increases from the extremities to the middle;
- in the middle area of the diaphysis, the equidistance is easily lost, the inner contours tend towards an elliptical shape, and the thickness of the cortical bone (in the same section) is not constant but increases in the anterior and posterior–lateral area of the tibia.
- the thickness of the cortical bone increases by more than 100% from the ends (where it connects with the cortical bone of the epiphyses) to the middle of the diaphysis.

A simplification of the modeling that has been adopted is that the two contours (interior and exterior) were considered equidistant in all 17 sections described above. Basically, the tendency to ovalize the inner contours in the sections from the middle of the tibia was neglected. However, in order to compensate for this, the dimensional increases of the cortical bone in the middle area of the diaphysis on the anterior and posterior–lateral direction were taken into account. This started at the ends with a thickness of 2.5 mm, which connect with the cortical bone of the epiphyses, with a continuous growth from the extremities to the middle of the diaphysis, resulting in the 3D model in Figure 3.

Figure 4 shows a section through 3D models of the cortical area of the epiphyses and diaphysis.

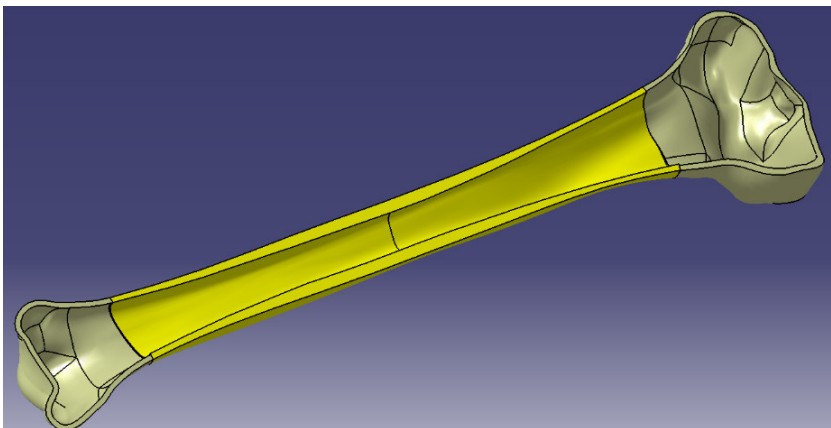

**Figure 4.** Section of human tibia—cortical bone.

Taking into account the real structure of the tibia, it should be noted that the trabecular bone area at the level of the epiphyses does not end abruptly, as shown in the models presented above, but decreases slowly towards the diaphysis, there is in this sense an area of interference between the epiphyses and diaphyses called the metaphysis. At the level of the metaphysis, the trabecular bone mass not only decreases quantitatively but there is also a decrease in its density.

Consequently, for the accuracy of the model, we considered it important to complete the modeled assembly with 2 more entities corresponding to the metaphyses (the red ones in Figure 5). These were modeled using the inner geometries placed towards the ends of the shaft, where the inner surfaces of the shaft became outer surfaces for the metaphyses. For the modeling of the inner surfaces of the metaphyses, two inner surfaces of revolution were generated in the form of ellipsoid parts to represent the passage from the medullary canal to the first trabecular formations.

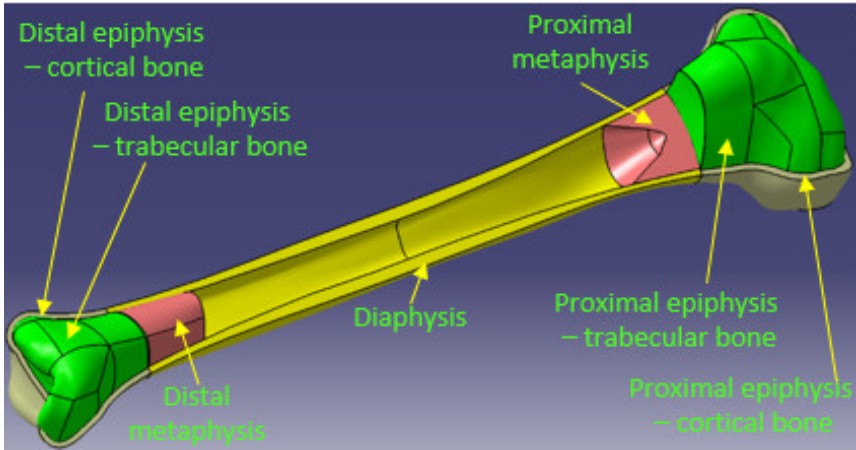

**Figure 5.** Section in human tibia—assembly of the entities.

Thus, the 5 entities modeled in CAD are: proximal epiphysis—cortical bone, proximal epiphysis—trabecular bone, proximal metaphysis, diaphysis, distal metaphysis, distal epiphysis—trabecular bone, distal epiphysis—cortical bone.

The final 3D model of the tibia that takes into account the actual bone structure is the result of assembling the entities presented above (Figure 5).

### 2.1.2. Virtual Simulation of the OWHTO

The modeling of the OWHTO was conducted taking into account the steps that are taken in the case of real surgery, with the highlighting (Figure 6a,b) of the geometric parameters that characterize the geometrical planning of this operation. Thus, the Center Of Rotation of Angulation Axis (CORA hereinafter), which is the hinge around which the creation of open wedge osteotomy takes place, is characterized by two parameters, V2, which represents the distance from the lateral tibial plateau to CORA, and V3, which represents the distance from the lateral cortex of the tibia to CORA (Figure 6a). Medial cortex (Figure 6b) is a curve located on the medial surface of the tibia, in a plane that intersects half of the tibial plateaus in the frontal plane, and cutting point (Figure 6a) is a point located on the medial cortex from which the initiation of the cut of the osteotomy plane takes place. The distance from this point to the tibial plateau V1 is also one of the parameters that characterize geometric planning. Finally, the fourth parameter V4 is the correction angle of the osteotomy wedge. CAD modeling of OWHTO surgery was performed parameterized so that it can be obtained from the generalized model how many customized models we want in order to optimize the geometric planning of the operation and obtain the best combination of parameters that ensure the correction angle in the best conditions.

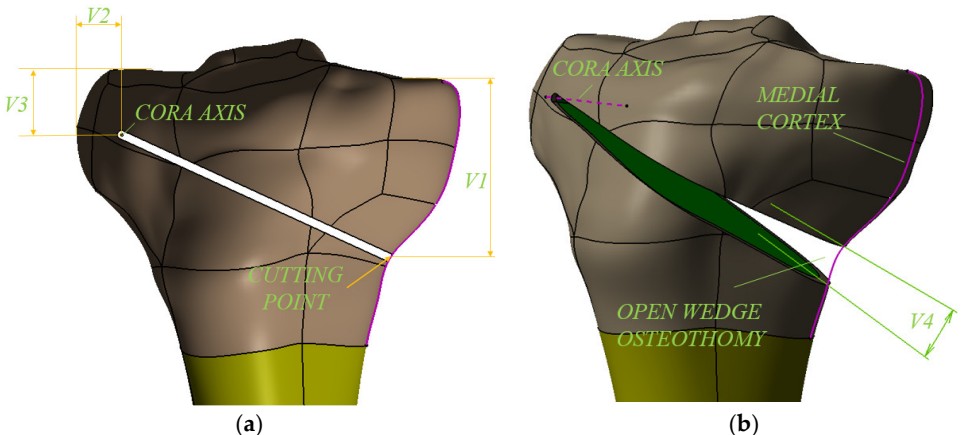

(**a**)             (**b**)

**Figure 6.** 3D modelling of the OWHTO—the parameters of the geometrical planning.

In order to complete the modeling of the OWHTO operation, CAD models were made for the 440.834S TomoFix osteosynthesis plate (standard) and the necessary fixing screws (Figure 7b), elements that preserve the obtained axial correction. The tibial model with the osteotomy pen created, the TomoFix plate and the screws were assembled according to the instruction on the surgical technique of the TomoFix Medial High Tibial Plate (DePuy Synthes; Synthes GmbH, Oberdorf, Switzerland) [33] resulting in the 3D assembly from Figure 7a. The CAD models made and presented will be used as geometric models in the CAE simulations in the Section 2.2.

### 2.2. FEM Analysis for Optimization of OWHTO Geometrical Planning

An important moment in the successful realization of the OWHTO is the creation of the correction angle, i.e., the creation of the osteotomy wedge (Figure 6b), after the realization of the cut of the osteotomy plane (Figure 6a). This can be achieved by elastic deformation of the bone with a specialized spacer [34]. In order to avoid microcracks that may appear in the CORA or in the articular area of the medial tibial plateau, it is essential to choose the correct lot of values for the 4 variables (V1, V2, V3, V4) that characterize the geometric planning shown in Figure 6.

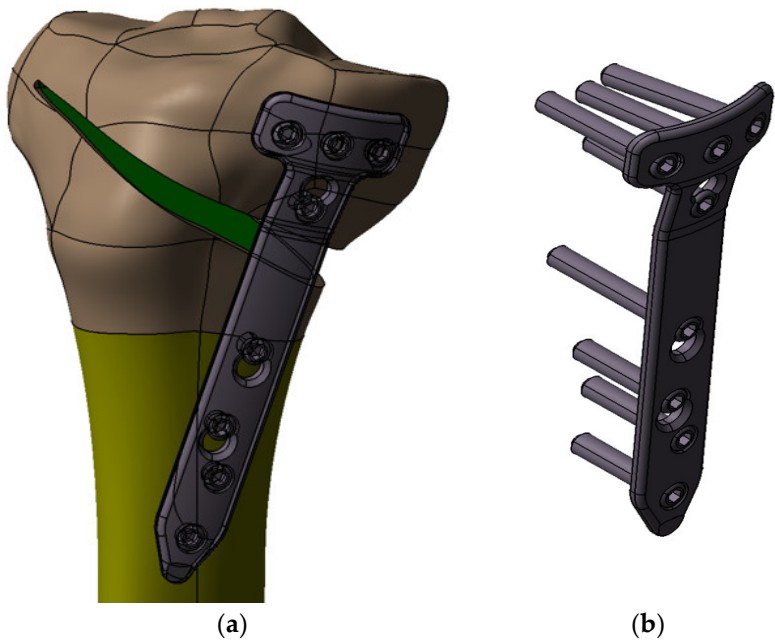

| (a) | (b) |
|-----|-----|

**Figure 7.** 3D modelling of the OWHTO with TomoFix Medial High Tibial Plate.

For the success of this operation and a fast and efficient recovery, an optimization of the 4 variables (V1, V2, V3, V4) is required. The optimization was performed using several sets of values and a research program was prepared using the Taguchi method, Table 1. The choice of variation levels for the 4 variables (minimum and maximum values) was made according to the surgical practice, the research and clinical trials performed. Thus, values between 30 mm and 50 mm are recommended for variable V1 [35,36]. A minimum level of 30 mm and a maximum of 40 mm were chosen for this variable. For CORA positioning, the recommended value ranges are 5–10 mm for V2 and 15–20 mm for V3 [35–37]. Consequently, for V2 the minimum level of 7.5 mm and the maximum level of 10 mm were chosen and for V3 the minimum level of 15 mm and the maximum level of 20 mm. The Taguchi method was also used to program the research, as it allows an efficient calculation of the average effects of the variables on the response sizes. The choice of the Taguchi method was also necessary due to the fact that it allows the observance of several restrictive conditions.

**Table 1.** Research design using the Taguchi method.

| Code | Values of Variables | | | |
|------|---------|---------|---------|---------|
|      | V1, mm | V2, mm | V3, mm | V4, ° |
| 1 | 30 | 7.5 | 15 | 6 |
| 2 | 30 | 7.5 | 15 | 12 |
| 3 | 30 | 10 | 20 | 6 |
| 4 | 30 | 10 | 20 | 12 |
| 5 | 40 | 7.5 | 20 | 6 |
| 6 | 40 | 7.5 | 20 | 12 |
| 7 | 40 | 10 | 15 | 6 |
| 8 | 40 | 10 | 15 | 12 |

OWHTO geometric planning optimization was conducted using finite element method (hereinafter FEM). Thus, by using FEM, the values of the following parameters were determined: equivalent stress, shear stress, displacement. The results obtained by FEM are response functions for the combinations of input sizes presented in Table 1 and allow the identification of those sets of values for which the three output sizes listed above will have the lowest values to avoid the appearance of microcracks in the tibia.

The known steps were taken to perform the FEM analysis. Thus, the geometric model developed in the CATIA V5R20 program and presented in Section 2.1 was imported as a geometric model using the STEP transfer standard and then discretized. Both modeled entities (cortical and cancellous) were considered to be homogeneous, isotropic and linearly elastic. Young's modulus for the tibia was 17 GPa for the cortical, 5 GPa for cancellous bones and Poisson's ratio was 0.33 [38–40]. The model was discretized (meshed) and its constraints and loads were applied (Figure 8a). As this analysis was aimed at the intraoperative creation of the osteotomy wedge, the value of the applied force was 175 N. The value was chosen taking into account the values existing in the literature between 120–200 N [36,41]. Figure 8b shows the meshed geometric model to which the constraints and loads were applied. As can be seen, since the area of interest is the proximal one of the tibia, a section was made in the middle of the tibia and only the upper part was analyzed. According to the CAD modeling described above, the tibia is a set of three entities with different mechanical characteristics: proximal epiphysis—cortical bone with Young's modulus 17 GPa, proximal epiphysis—trabecular bone with Young's modulus 5 GPa and proximal metaphysis with Young's modulus 1 GPa [38–40]. For TomoFix plate the Young's modulus was 110 GPa.

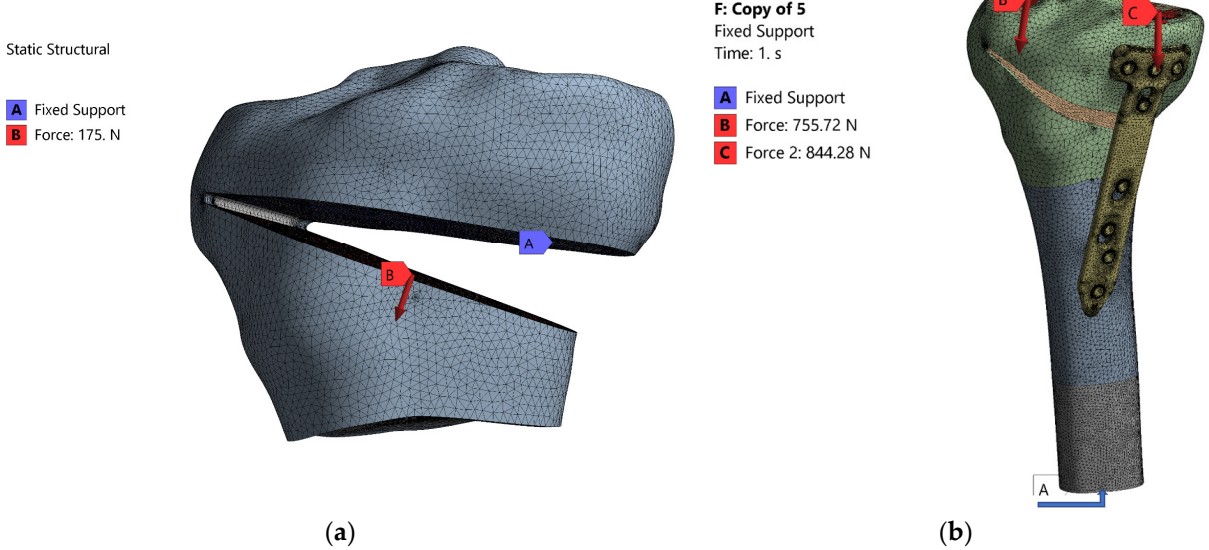

(**a**)  (**b**)

**Figure 8.** Meshing the models (**a**) and application of stresses and constraints (**b**).

Due to the geometrical complexity of the model, for the meshing process we used tetrahedral elements with 4 nodes. In order to improve the precision of the mesh we established a maximum edge length of the tetrahedral element at 1.5 mm and a minimum edge length of $2.4 \times 10^{-3}$ mm. These settings led to 568,741 elements, connected in 794,926 nodes for each geometrical model.

Following the FEM analysis performed with the help of the ANSYS program, the values for equivalent stress, shear stress, displacement were determined.

### 2.3. Statistical Analysis of Research Results

Following the application of the Taguchi method for the design of the research, results obtained were statistically processed using ANOVA analysis. Thus, the types of variables considered were initially established, namely:

- influencing factors (V1, V2, V3, V4—independent variables);
- system responses (equivalent stress, shear stress, displacement—dependent variables whose values need to be optimized).

Under these conditions, the aim was to determine the sets of values for V1, V2, V3, V4 so that the equivalent stress, shear stress, displacement have minimum values. Additionally,

within the statistical analysis, the aim was to establish mathematical models that would offer the dependence of the system response (equivalent stress, shear stress, displacement) depending on the influencing factors: (V1, V2, V3, V4). Establishing these mathematical models also allows the identification of other sets of values for system responses and for other sets of values of influence factors other than those established using the Taguchi method (Table 1). In order to determine whether the determined models describe very well the phenomena analyzed in the next stage, the values of correlation coefficient ($R^2$) and the adjusted correlation coefficient ($R^2$ adj) were calculated. Thus, the closer the values of $R^2$ and $R^2$ adj are to the value 1, the better the mathematical models describe the phenomena for which they were established.

Checking the adequacy of mathematical models and influencing factors is significant for the model, and was also performed by establishing the values of F and p, respectively. If F has high values and p has low values ($p < 0.05$), this indicates that the model is adequate and the variable corresponding is very significant. Moreover, in order to verify which of the factors (V1, V2, V3, V4) influence the system responses the most, a multiple regression analysis was performed that allowed the values to be established for V1,b*,V2,b*,V3,b*,V4,b*. Thus, the highest value of V1,b*,V2,b*,V3,b*,V4,b * demonstrates that the respective influencing factor has the greatest influence on the system's responses.

The statistical processing of the obtained results and the establishment of the previously mentioned parameters provide a complete picture of the studied phenomena and can provide information for other values of the influencing factors apart from the analyzed ones.

### 2.4. The Numerical Simulation of the Postoperative Behavior of the Tibia Plate Ensemble TomoFix of Osteosynthesis

The second numerical analysis performed targeted the whole tibia—osteosinthesys plate after surgery. Specifically, the condition of stresses and deformities in the tibia and osteosynthesis plaque was studied as a function of positioning the point of intersection between the medial–lateral articular line and the corrected mechanical axis. As mentioned in the Introduction Section, this point can be placed in the middle of the knee (normal physiological case), but, in general, an overcorrection of up to 25% to the side, measured from the middle of the joint, is preferred. Obviously, changing this position affects the geometric planning of the surgery.

Figure 9 shows the loads on the tibial plateaus of a load of 1600 N consisting of physiological loads—1400 N and surgical loads—200 N [10–13], and Table 2 shows how the load transfer from the medial takes place to the side with the change in the position of the mechanical axis.

**Table 2.** The load values depending on the change in the position of the mechanical axis.

| Mechanical Axis Position Change between 50–75% (Lateral–Medial) % | Changing the Position of the Mechanical Axis in mm (Lateral–Medial) mm | Medial–Lateral Load Transfer in N/mm | Loading in the Medial Area, N | Loading in the Lateral Area, N | Total Load, N |
|---|---|---|---|---|---|
| 0 (50%) | 0 (CENTER) | 0 | 960 | 640 | 1600 |
| 1 | 0.78 | 31.98 | 937.02 | 662.98 | 1600 |
| 2 | 1.56 | 49.89 | 919.11 | 680.89 | 1600 |
| 3 | 2.34 | 74.83 | 894.17 | 705.83 | 1600 |
| 4 | 3.12 | 99.78 | 869.22 | 730.78 | 1600 |
| 5 | 3.9 | 124.72 | 844.28 | 755.72 | 1600 |
| 6 | 4.68 | 149.67 | 819.33 | 780.67 | 1600 |
| 7 | 5.46 | 174.61 | 794.39 | 805.61 | 1600 |
| 8 | 6.24 | 199.56 | 769.44 | 830.56 | 1600 |
| 9 | 7.02 | 224.50 | 744.50 | 855.50 | 1600 |

**Table 2.** *Cont.*

| Mechanical Axis Position Change between 50–75% (Lateral–Medial) % | Changing the Position of the Mechanical Axis in mm (Lateral–Medial) mm | Medial–Lateral Load Transfer in N/mm | Loading in the Medial Area, N | Loading in the Lateral Area, N | Total Load, N |
|---|---|---|---|---|---|
| 10 | 7.8 | 249.44 | 719.56 | 880.44 | 1600 |
| 11 | 8.58 | 274.39 | 694.61 | 905.39 | 1600 |
| 12 | 9.36 | 299.33 | 669.67 | 930.33 | 1600 |
| 12.5 | 9.75 | 311.81 | 657.20 | 942.81 | 1600 |
| 13 | 10.14 | 324.28 | 644.72 | 955.28 | 1600 |
| 14 | 10.92 | 349.22 | 619.78 | 980.22 | 1600 |
| 15 | 11.7 | 374.17 | 594.83 | 1005.17 | 1600 |
| 16 | 12.48 | 399.11 | 569.89 | 1030.11 | 1600 |
| 17 | 13.26 | 424.05 | 544.95 | 1055.05 | 1600 |
| 18 | 14.04 | 449.00 | 520.00 | 1080.00 | 1600 |
| 19 | 14.82 | 473.94 | 495.06 | 1104.94 | 1600 |
| 20 | 15.6 | 498.89 | 470.11 | 1129.89 | 1600 |
| 21 | 16.38 | 523.83 | 445.17 | 1154.83 | 1600 |
| 22 | 17.16 | 548.78 | 420.22 | 1179.78 | 1600 |
| 23 | 17.94 | 573.72 | 395.28 | 1204.72 | 1600 |
| 24 | 18.72 | 598.67 | 370.33 | 1229.67 | 1600 |
| 25 | 19.5 | 623.61 | 345.39 | 1254.61 | 1600 |

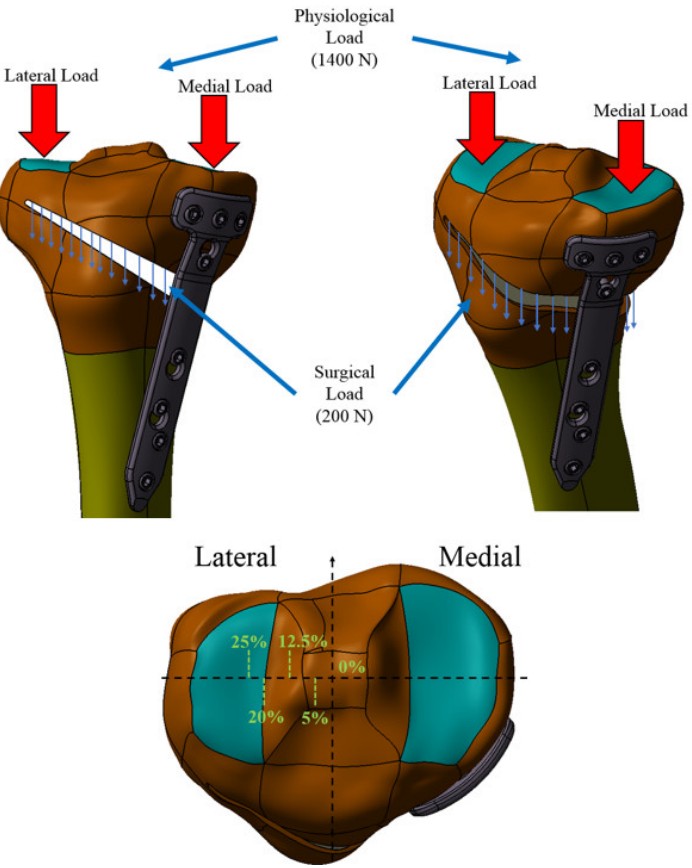

**Figure 9.** Loads on the ankle joint.

When the mechanical axis intersects the center of the joint, the load is the known: 60% medial and 40% lateral. With there is a change in this position by 1 mm, there is a redistribution of medial–lateral loads by 41 N [27]; thus, being able to calculate the medial

and lateral loads for each percentage position of the axis between 0% (joint center) and 25% position external overcorrection to the side. Table 2 shows the values of the loads according to the change in the position of the mechanical axis. The second column of Table 2 shows the values in millimeters for each percentage position in the case of the CAD model made in the previous subsection.

Numerical analysis was performed using the ANSYS program for 5 of the possible cases of modification of the mechanical axis given by overcorrections most often used [18–26]), namely, for the situation where the mechanical axis is arranged in the middle area of the knee, and then the axis modification mechanical is 0%, 5%, 10%, 12.5%, 20% and 25%, respectively.

It should be noted, with regard to the medial–lateral load transfer that, in the 12.5% Fujisawa Point situation, there is practically a reversal of the normal load, in the sense that the lateral plate is loaded approximately 60% and the lateral one 40%. Additionally, noteworthy are the large and very large loads of the lateral tibial plateau for the positions of 20% and 25%, respectively (Figure 10).

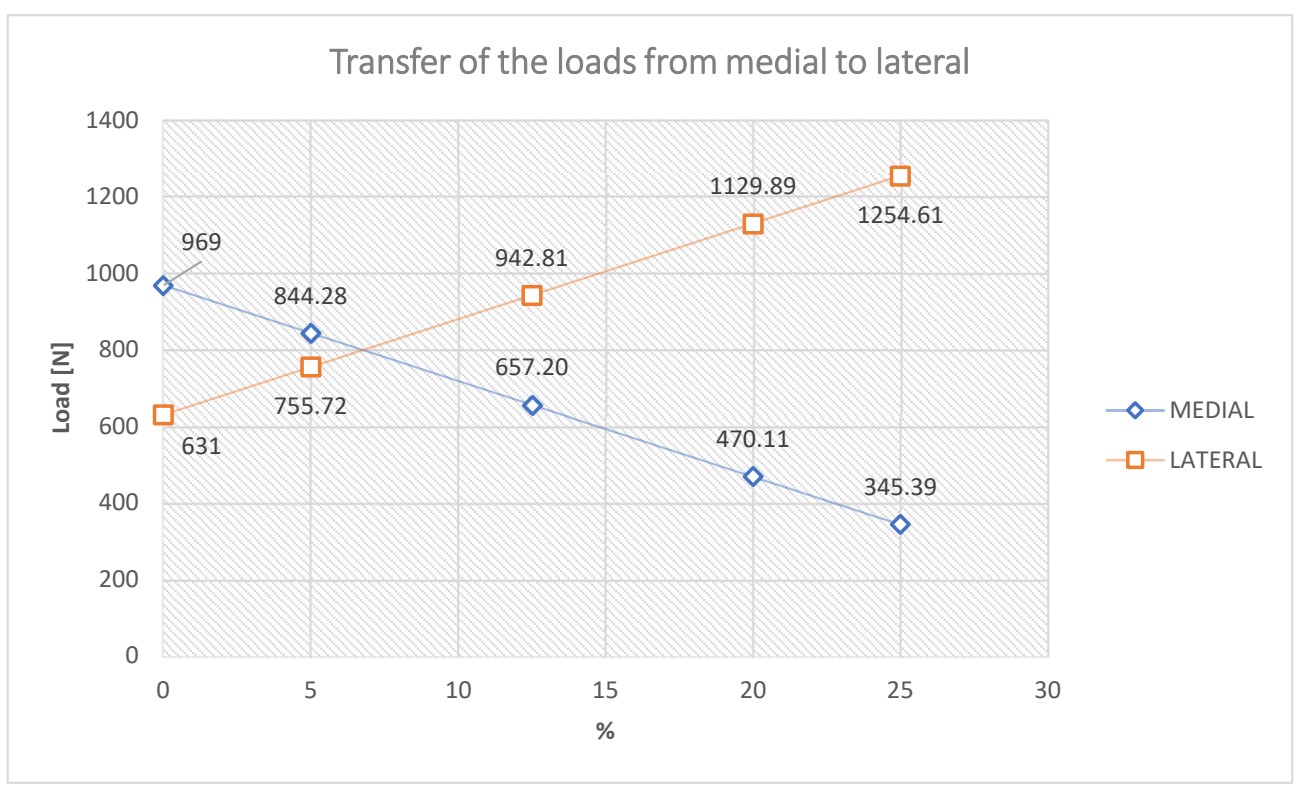

**Figure 10.** Transfer of the load medial–lateral tibial plateau.

Equivalent stress, shear stress, displacement were evaluated for each of the 5 positions in order to determine the position that provides the best possible postoperative stability of the knee.

## 3. Results and Discussion

The results of the numerical analyses are presented next on the two research directions mentioned, namely, the optimization of the geometric parameters of the OWHTO in the intraoperative realization of the osteotomy wedge, and the postoperative behavior of the tibia–TomoFix osteosynthesis flat existing axial, preoperatively. For good reproducibility of the obtained results they were statistically processed.

### 3.1. OWHTO Geometric Planning Optimization

The osteotomy wedge during the OWHTO should be carried out after careful geometric planning because its size and positioning depends on the purpose of the intervention with

the achievement of the proposed goal: to eliminate the axial deviation of the mechanical axis of the foot affected by OA. The four established variables (V1, V2, V3, V4), (Figure 6a,b) characterize geometrically the osteotomy wedge completely.

The results of the numerical analyses of equivalent stress, shear stress and displacement were recorded for each combination of levels of variation in the variables, taking into account the experimental design obtained by the Taguchi method. These are presented in Table 3. Obviously, the optimal criterion followed is that in which the values of the parameters of the three response functions (equivalent stress, shear stress and displacement) are minimal.

**Table 3.** Parameter values (equivalent stress, shear stress, displacement) obtained by applying FEM.

| Code | V1 | V2 | V3 | V4 | Equivalent Stress | Shear Stress | Displacement |
|------|-----|-----|-----|-----|------|------|------|
| 1 | 30 | 7.5 | 15 | 6 | 168.65 | 90.77 | 1.18 |
| 2 | 30 | 7.5 | 15 | 12 | 170.62 | 90.33 | 1.14 |
| 3 | 30 | 10 | 20 | 6 | 99.6 | 53.23 | 0.6 |
| 4 | 30 | 10 | 20 | 12 | 97.13 | 51.28 | 0.58 |
| 5 | 40 | 7.5 | 20 | 6 | 185.25 | 97.91 | 1.34 |
| 6 | 40 | 7.5 | 20 | 12 | 199.29 | 104.84 | 1.48 |
| 7 | 40 | 10 | 15 | 6 | 99.91 | 52.09 | 0.58 |
| 8 | 40 | 10 | 15 | 12 | 105.2 | 55.07 | 0.64 |

A first observation is that for all three response functions, the values domain of the results obtained is quite large. Thus, the maximum values in relation to the minimum are approximately twice as high, and this demonstrates the fact that an improper choice of the values of V1, V2, V3, V4 can cause a doubling of the stresses and deformations in the CORA area of the tibia subjected to angular correction.

From the analysis of the results presented in Table 3, the importance of the geometric positioning of CORA was noticed, especially with respect to the lateral cortex of the tibia (V2). In all cases where V2 has a minimum value of 7.5 mm, there are very high values of equivalent stress but also of the other two response functions. It follows that lower values for this variable (5 mm), recommended in some research [35–37], will produce equivalent and higher voltages, which can lead to microcracks in the CORA area.

Even for a V2 equal to 7.5 mm, it can be seen that, in the case of the combination of parameters from code 6, a value of the equivalent stress is obtained very close to the ultimate compression tensile value for the human tibia, which is around 200 MPa [40].

Combination 6 is the most unfavorable in terms of stress and strain states in CORA for angular corrections higher than 12°, and combination 5 for small correction angles of 6°. At the opposite pole are the combinations of codes 4 (12° correction angle) and 7 (6° correction angle), for which all three response functions have the lowest values.

Figures 11 and 12 show the distributions of the equivalent stress, shear stress and displacement for codes 6 and 7.

It is also worth noting that, although some research [28,38] optimizes these parameters, the combinations between them are often limited. The results obtained in the research demonstrate the need and importance of optimizing all geometric parameters and completing some previous results, which took into account, for example, only two parameters, V1 and V4 [28].

In order to be able to observe the influence that the four stress and displacement variables have, a statistical processing of the obtained results was also performed. Thus, a regression analysis was initially performed and the regression equations presented in Table 4 were obtained.

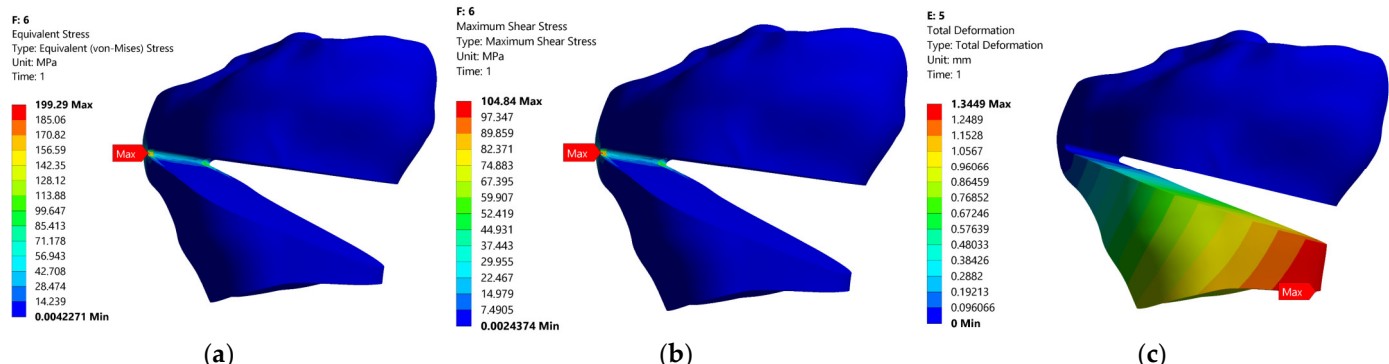

**Figure 11.** Static FEM analysis for model with code 6: (**a**)—equivalent stress, (**b**)—shear stress, (**c**)—displacement.

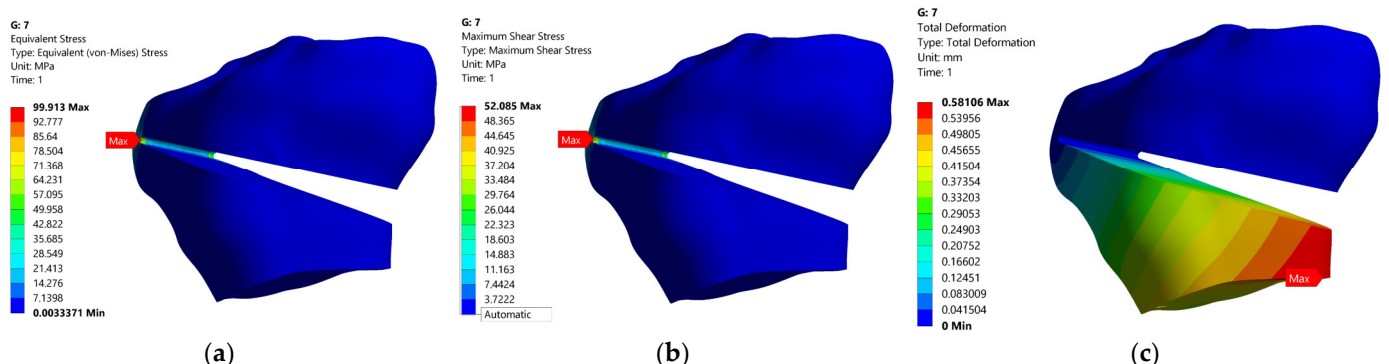

**Figure 12.** Static FEM analysis for model with code 7: (**a**)—equivalent stress, (**b**)—shear stress, (**c**)—displacement.

**Table 4.** Regression equations for: equivalent stress, shear stress, displacement.

| Parameters | Regression Equation |
|---|---|
| Equivalent stress | Equivalent stress = 336.1 + 1.341 V1 − 32.20 V2 + 1.844 V3 + 0.785 V4 |
| Shear stress | Shear stress = 184.4 + 0.608 V1 − 17.218 V2 + 0.950 V3 + 0.313 V4 |
| Displacement | Displacement = 2.413 + 0.01350 V1 − 0.2740 V2 + 0.02300 V3 + 0.00583 V4 |

In order to observe whether the models presented in Table 4 are relevant, in the next step the correlation coefficient ($R^2$) and the adjusted correlation coefficient ($R^2$ adj) were established. The values of these coefficients suggest whether the model provides a satisfactory representation of the process. Thus, for a model to be satisfactory for a process, $R^2$ and $R^2$ adj must be at least 0.80. The values obtained for $R^2$ and $R^2$ adj, respectively, for the three models in Table 4 are presented in Table 5.

**Table 5.** The values of $R^2$ and $R^2$ adj, respectively, for the regression equations.

| Parameter Regression Equations | $R^2$ | $R^2$ adj |
|---|---|---|
| Equivalent stress | 0.99461817 | 0.98744240 |
| Shear stress | 0.99393486 | 0.98584801 |
| Displacement | 0.98998964 | 0.97664250 |

The results presented in Table 5 show that both the correlation coefficient ($R^2$) and the adjusted correlation coefficient ($R^2$ adj) have values much higher than 0.8, very close

to 1, which demonstrates that the models in the equations of regression have a very good representation of the process.

Additionally, in order to observe if the models represented by the regression equations are significant, the values of F and p were calculated. If the value of F is high, it suggests that the model is significant, and if the value of p is less than 0.05, the model is significant. The values calculated for F and p are shown in Table 6.

**Table 6.** The values of F and p, respectively, for the regression equations.

| Parameter Regression Equations | F | p |
|---|---|---|
| Equivalent stress | 138.6078 | 0.000984 |
| Shear stress | 122.9075 | 0.001177 |
| Displacement | 74.17242 | 0.0026 |

The analysis of the respective values of F and p presented in Table 6 show that F has high enough values and p has low enough values for the models given by the regression equations and the meaning of their terms to be significant. However, the displacement model is less significant compared to the other two, but it is still significant enough considering the values of F and p. It is also very important to know the influence that each of the four variables has on the three parameters and, thus, a response for signal to noise ratios (smaller is better) was established, see Table 7, but a multiple regression analysis was also performed, which allowed the values for V1,b*,V2,b*,V3,b*,V4,b* to be established, see Table 8.

**Table 7.** Response for signal to noise ratios (smaller is better).

| Level | V1 | V2 | V3 | V4 |
|---|---|---|---|---|
| 1 | −42.22 | −45.13 | −42.40 | −42.46 |
| 2 | −42.94 | −40.04 | −42.76 | −42.70 |
| Delta | 0.72 | 5.10 | 0.36 | 0.24 |
| Rank | 2 | 1 | 3 | 4 |

**Table 8.** The values of V1,b*,V2,b*,V3,b*,V4,b* for regression equations.

| Parameter Regression Equations | V1,b* | V2,b* | V3,b* | V4,b* |
|---|---|---|---|---|
| Equivalent stress | 0.163 | 0.98 | 0.112 | 0.057 |
| Shear stress | 0.138 | 0.98 | 0.108 | 0.043 |
| Displacement | 0.190 | 0.96 | 0.162 | 0.049 |
| Rank | 2 | 1 | 3 | 4 |

From the analysis of the data presented in Tables 7 and 8, respectively, it could be observed that, both after establishing the response for signal to noise ratios and after establishing the values for V1,b*,V2,b*,V3,b*,V4,b*, the same ranking of the influence of the four variables was obtained (V1, V2, V3, V4). Thus, the variable V2 has the greatest influence, which demonstrates that, in practice, it is very important to pay special attention to the positioning of the CORA in relation to the lateral cortex of the bone. It is also worth noting that the influence of the correction angle has a small influence on the stresses and strains in the CORA compared to the other parameters and highlights the importance of the geometric planning of the OWHTO. It is worth noting the usefulness of the computer-aided approach to CAD-CAM methods of medical aspects through the superior possibilities of processing the results obtained on virtual models. Such biomechanical approaches are present in the literature [41,42].

### 3.2. Postoperative Behavior of the Tibia Plaque Ensemble TomoFix by Osteosynthesis Taking into Account the Overcorrection Degree of the Mechanical Axis of the Leg

The main purpose of the OWHTO is to eliminate the axial deviation that occurs at the level of the mechanical axis of the tongue, but, for a favorable postoperative evolution and a very good recovery, overcorrections are made regarding the position of this axis. In this sense, there are experimental researches or clinical studies [18–26] that recommend different values or sets of values for these overcorrections. The results we obtained in this study due to the computer-assisted approach allow a continuous assessment of the values of stress and strain that occur in the bone and the TomoFix plate for the entire range of overcorrection, from the middle joint to a maximum value of 25%, towards the medial part of the tibia.

Table 9 shows the maximum total deformations and the maximum equivalent stresses for the main five positions of the corrected mechanical axis (0%, 5%, 12.5%, 20%, 25%). Figures 13–15 show the distribution of total deformations (a) of equivalent stress states in the bone (b), and in the flat TomoFix (c) for three distinct positions: 0% overcorrection (Figure 13), Fujisawa Point 12.5% and the maximum overcorrection of 25%.

**Table 9.** Maximum values for deformations, bone stress equivalent, plaque stress equivalent obtained after FEM analysis.

| Modification of the Position of the Mechanical Axis, % | 0 | 5 | 12.5 | 20 | 25 |
|---|---|---|---|---|---|
| Total Deformation, mm | 0.409 | 0.344 | 0.452 | 0.627 | 0.772 |
| Equivalent Stress in bone, MPa | 115.56 | 114.83 | 113.7 | 112.57 | 111.82 |
| Equivalent Stress in the plate, MPa | 201.47 | 182.07 | 152.85 | 123.62 | 111.51 |

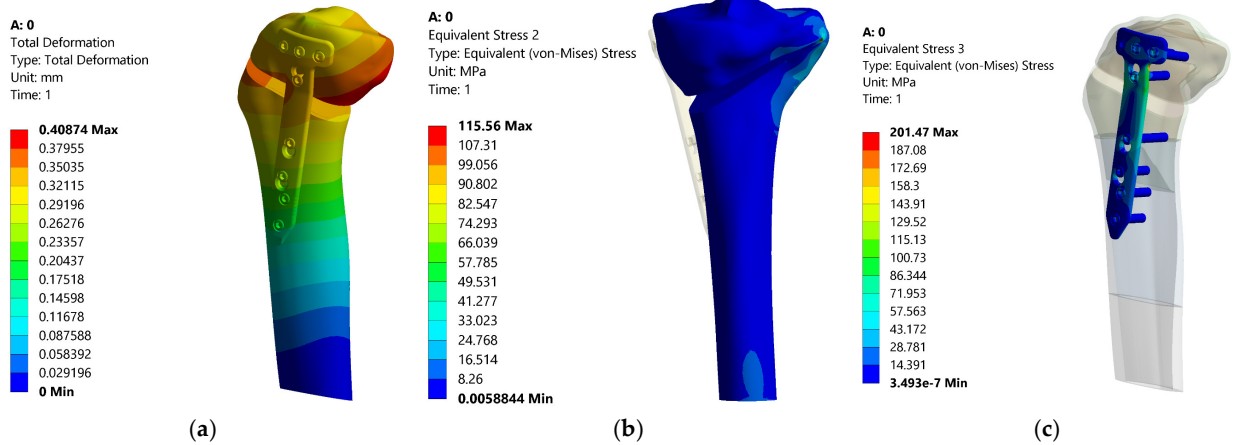

**Figure 13.** Distributions of the total deformation, equivalent stress in tibia and equivalent stress in the plate—0%.

Figures 16–18 show the variations in the three response functions studied depending on the degree of overcorrection of the mechanical axis.

Analyzing the presented results, a first observation would be that there is a substantial change in the values of deformities for the equivalent stress in the plate and a smaller change in the equivalent stress in the tibia.

This first aspect reveals that the fixation of the TomoFix plate is a stable one because an almost constant equivalent stress can be noticed in the tibia bone regardless of the degree of overcorrection (Figure 17). The variation is a fairly small linear one between a maximum value of 115.56 MPa for 0% (i.e., without overcorrection) and a minimum of 111.82 MPa for a maximum overcorrection of 25%. The results thus confirm the research in [43], which highlights the superiority and stability of TomoFix flat fixation relative to other fastening systems.

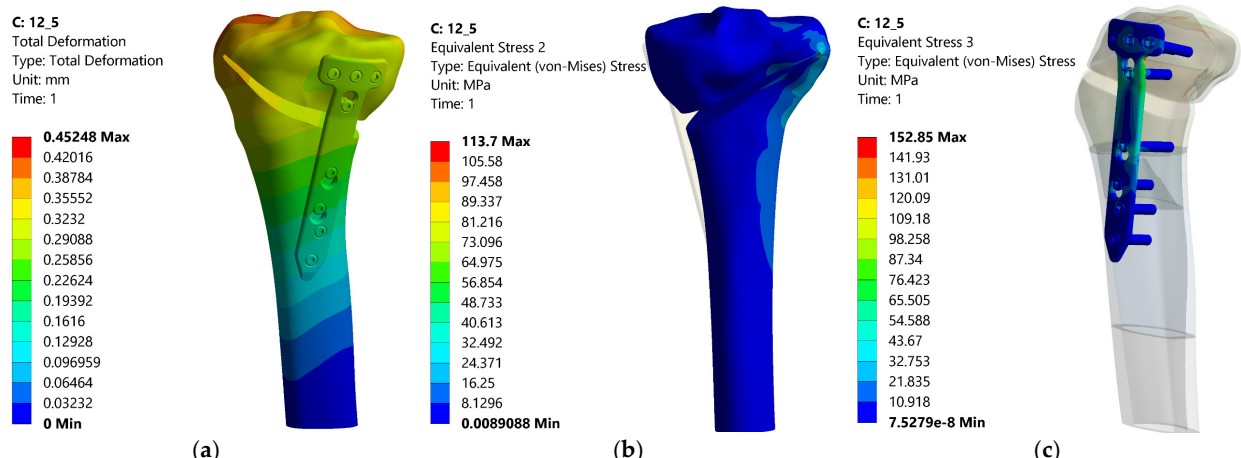

**Figure 14.** Distributions of the total deformation, equivalent stress in tibia and equivalent stress in the plate—12.5%.

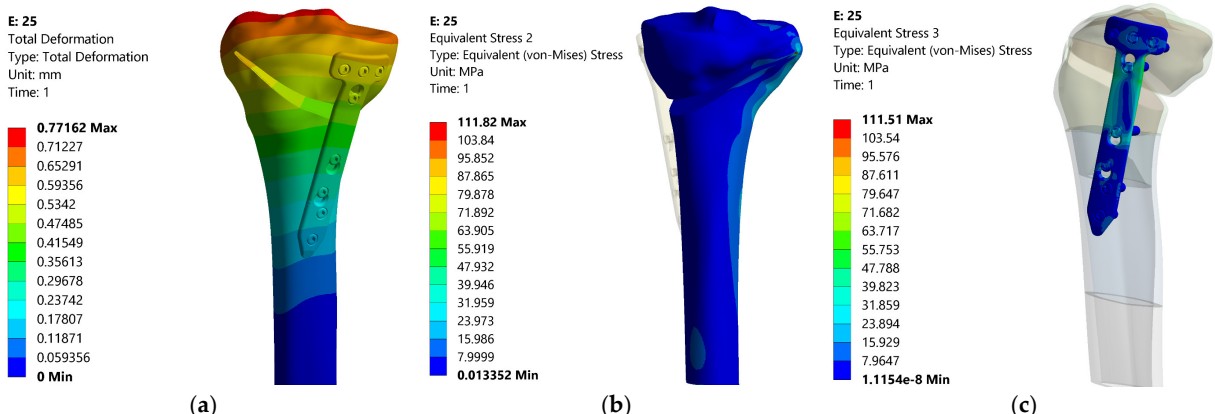

**Figure 15.** Distributions of the total deformation, equivalent stress in tibia and equivalent stress in the plate—25%.

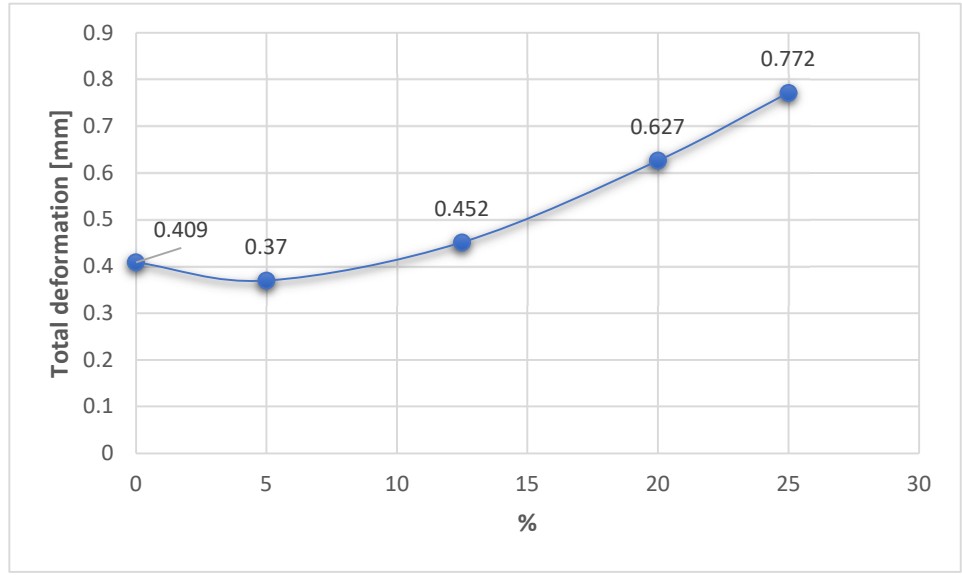

**Figure 16.** Variation in the total deformation related to the overcorrection positions.

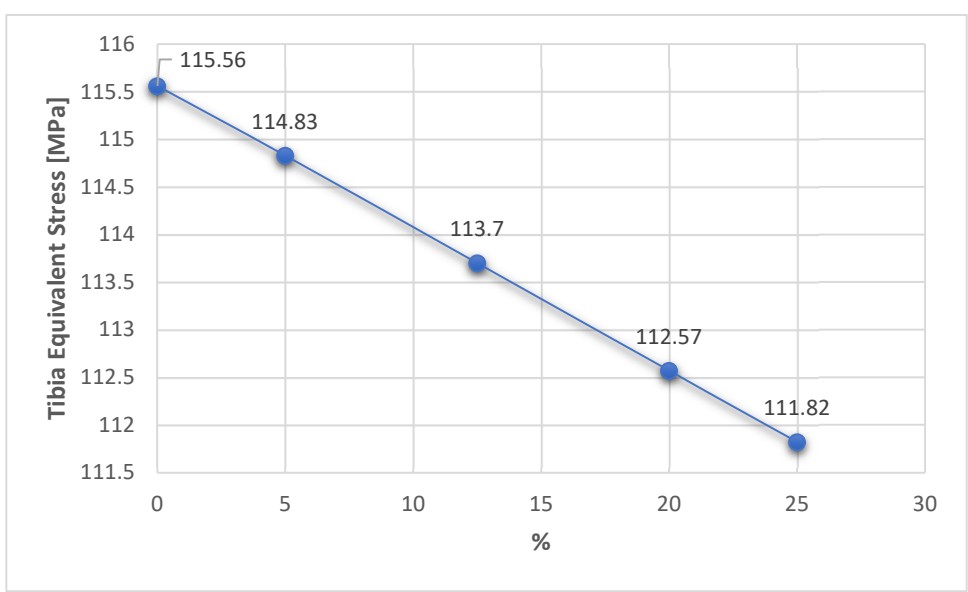

**Figure 17.** Variation in the tibia equivalent stress related to the overcorrection positions.

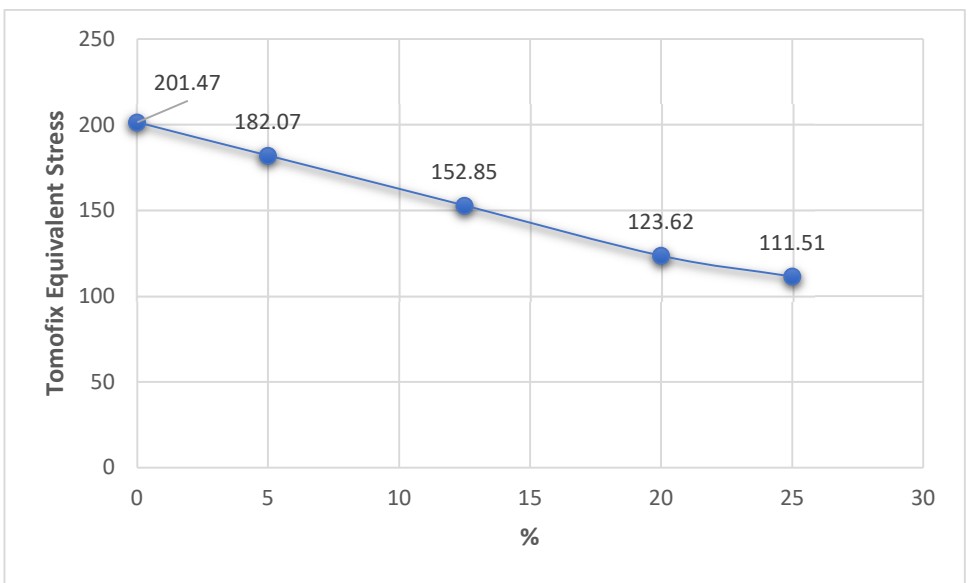

**Figure 18.** Variation in the TomoFix equivalent stress related to the overcorrection positions.

In the case of the TomoFix equivalent stresses, there is a more consistent, nonlinear variation of about 100 MPa but with the same tendency to decrease from 0% to 25% (Figure 18). It is also worth noting that for a 25% change in the position of the mechanical axis, an approximately equal distribution of stress is obtained in both the plaque and the bone, but with an increase in the total deformation of about 100% compared to the situation in which the change in the position of the mechanical shaft was only 5%.

In fact, with regard to total deformation, it can be seen that for overcorrections of up to 12.5% (Fujisawa Point), the values for total deformation are reasonable, grouped around 0.4 mm, then having a significant increase for large overcorrections (20–25%). The result confirms the research in the papers [18,25,26], which emphasized that excessive overcorrection is not recommended as it can lead to wear of the cartilage from the lateral compartment and other degenerations and dysfunctionalities.

Analyzing all these results, it can be stated that the optimum point, taking into account the three response functions analyzed, is the Fujisawa Point, which has equivalent reasonable stresses of 113.7 MPa in the tibia bone, 152.85 MPa in the TomoFix plate under

the condition of a total deformation of 0.452 mm, quite close to the resulting minimum value. This is a numerical confirmation of the experimental results in the article [44].

Regarding the strengths and limitations of this research, it should be noted that the use of CAD models that take into account the mechanical characteristics and actual structure of human bones may provide a closer assessment of reality compared to possible experimental assessments performed on bovine or porcine bones. It is also possible to analyze the influence of variables that influence the geometric planning of the OWHTO by simulating loads and constraints on the same unique CAD bone model. It is known to limit the importance of experimental methods on real bones, given the impossibility of performing a considerable number of absolutely identical experimental samples.

CAD-FEM modeling also proves its usefulness in establishing its optimal position as the point of intersection between the medial–lateral articular line and the corrected mechanical axis. Strengths in this regard would be the accuracy with which loads can be made on the surfaces of the medial and lateral plates of the tibia 3D model and the possibility of accurately adjusting (at any point on the joint line) the load transfer from medial to lateral specific to OWHTO correction.

Obviously, the approach has some limitations, such as, first of all, the fact that the study is performed on a CAD model, which, no matter how well it is performed, cannot be 100% identical to the real human bone. The effect of soft tissues (muscles, cartilage, meniscus) is also neglected.

Overall, we consider that the strengths highlighted above provide a relevant assessment of the objectives proposed in the paper and with a good degree of generality.

### 4. Conclusions

The utility of computer-assisted methods offers the possibility of the virtual creation of a very large number of situations, models or simulations that allow the exhaustive analysis of the studied phenomenon. The creation of CAD models using real bone structure has led to an increase in the accuracy and correctness of EMF assessments. The use of the Taguchi method of planning numerical evaluations allowed an efficient calculation of the average effects of the variables on the response sizes and the observance of several restrictive conditions. The applicability of the modeling is revealed by finding optimal combinations of the parameters that define the geometric planning of the OWHTO as well as highlighting the most unfavorable combinations, both for small correction angles, 6°, and for large angles, 12°. The superiority of the Fujisawa Point for the correction of axial deviations resulting from OA is also confirmed by the computer-assisted methods used. We consider that the results obtained can be benchmarks for orthopedic doctors to establish the geometric parameters that characterize the planning of the operation, but also for choosing the optimal overcorrection for the corrected mechanical axis of the tongue in order to obtain a good stability of the joint and consolidation of the osteotomy gap, and improve the patient's quality of life.

**Author Contributions:** Conceptualization, N.F.C., I.I.C., R.D.D., A.H. and S.R.F.; methodology, N.F.C., R.D.D., I.I.C. and M.O.; software, N.F.C., I.I.C. and A.H.B.; validation, N.F.C., I.I.C. and M.O.; formal analysis, N.F.C. and A.H.B.; investigation, A.H., R.D.D., M.O. and S.R.F.; resources, N.F.C., M.D.R., I.I.C., M.O. and S.R.F.; data curation, N.F.C., M.D.R. and M.O.; writing—original draft preparation, N.F.C., I.I.C., A.H. and A.H.B.; writing—review and editing, N.F.C., M.D.R., I.I.C. and M.O.; visualization, N.F.C. and M.O.; supervision, N.F.C. and M.O.; project administration, N.F.C. All authors have read and agreed to the published version of the manuscript.

**Funding:** This research received no external funding.

**Informed Consent Statement:** Not applicable.

**Data Availability Statement:** Not applicable.

**Conflicts of Interest:** The authors declare no conflict of interest.

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
