# Peer review of "A Computer-Assisted Approach Regarding the Optimization of the Geometrical Planning of Medial Opening Wedge High Tibial Osteotomy"

_applsci, doi:10.3390/app12136636_

Round 1

Reviewer 1 Report

Dear authors,

I read your article with interest but I believe that extensive changes are necessary for the article to be accepted in this magazine. Here are my suggestions

- Introduction

This section should be extensively reviewed and perhaps summarized by removing some repetitive statements that are not relevant to the purpose of the article.

Line 56: add epidemiological data on knee osteoarthritis and briefly also some data on conservative treatment. In this regard it might be useful to mention the following articles:

 https://doi.org/10.3390/app11188711

Line 67-121: these paragraphs are very repetitive. It is not clear what the purpose or aims of the study are and there are too many repetitions. Better clarify the purpose or purposes of the study without going too far and add additional information previously in the introduction.

Line 85: add a bibliographic reference

- Methods

This paragraph should also be thoroughly revised because in my opinion it is very confusing and could bore the reader. If possible it would be appropriate to summarize this paragraph.

Line 234: why do you present another purpose of the study here if the aims of the study have already been expressed in the final part of the introduction section ???

I believe that a subsection is missing in which the statistical analysis used in the study is explained. To add.

Line 133: add a bibliographic reference

- Results

Add the strengths and limitations of this research

Reviewer 2 Report

Dear Authors, congratulations on your interesting work. Overall, I consider the article to be correct and carefully prepared.

- why a rectangular mesh was used for numerical tests, and not a triangular mesh,

- Figure 10 - for (,) please insert (.)

- The authors summarize the problems, but do not provide possible solutions, hence the doubts as to the practical application of the model. Therefore, I propose to improve this part of the work.

Round 2

Reviewer 1 Report

Dear authors, the corrections have been correctly carried out